# Resection of Intramedullary Hemangioblastoma: Timing of Surgery and Its Impact on Neurological Outcome and Quality of Life

**DOI:** 10.3390/medicina59091611

**Published:** 2023-09-06

**Authors:** Michael Schwake, Sarah Ricchizzi, Sophia Krahwinkel, Emanuele Maragno, Stephanie Schipmann, Walter Stummer, Marco Gallus, Markus Holling

**Affiliations:** 1Department of Neurosurgery, University Hospital Münster, D-48149 Münster, Germany; s_ricc01@uni-muenster.de (S.R.); sophia.krahwinkel@ukmuenster.de (S.K.); emanuele.maragno@ukmuenster.de (E.M.); stephanie.schipmann@ukmuenster.de (S.S.); walter.stummer@ukmuenster.de (W.S.); marco.gallus@ukmuenster.de (M.G.); markus.holling@ukmuenster.de (M.H.); 2Department of Neurosurgery, University of Bergen, N-5020 Bergen, Norway

**Keywords:** von Hippel-Lindau, intramedullary hemangioblastoma, spinal intramedullary tumors, minimally invasive spine surgery, fluorescence-guided resection

## Abstract

*Background and Objectives*: Spinal intramedullary hemangioblastomas (SIMH) are benign vascular lesions that are pathological hallmarks of von Hippel-Lindau disease (vHL) and constitute the third most common intramedullary neoplasm in adults. So far, maximal and safe resection is the first choice of treatment. However, as SIMH show no malignant transformation, it remains unclear whether surgical resection is beneficial for all patients. *Materials and Methods*: We retrospectively analyzed the surgical outcomes of 27 patients who were treated between 2014 and 2022 at our neurosurgical department and investigated potential risk factors that influence the surgical outcome. Pre- and postoperative neurological status were classified according to the McCormick scale. Furthermore, surgical quality indicators, such as length of hospital stay (LOS; days), 90-day readmissions, nosocomial infections, and potential risk factors that might influence the surgical outcome, such as tumor size and surgical approach, have been analyzed. In addition to that, patients were asked to fill out the EQ-5D-3L questionnaire to assess their quality of life after surgery. *Results*: Surgery on SIMH patients that display no or minor neurological deficits (McCormick scale I or II) is associated with a favorable postoperative outcome and overall higher quality of life compared to those patients that already suffer from severe neurological deficits (McCormick scale III or IV). *Conclusion*: Early surgical intervention prior to the development of severe neurological deficits may offer a better neurological outcome and quality of life.

## 1. Introduction

Spinal intramedullary hemangioblastomas (SIMH) represent WHO grade 1 benign vascular lesions that are one of the pathological hallmarks of von Hippel-Lindau disease (vHL) [1,2,3,4]. Clinical presentation is related to intramedullary tumor location and patients usually display pain and/or sensorimotor deficits [5]. Furthermore, a few cases were reported where spinal hemangioblastomas caused a subarachnoid hemorrhage or hematomyelia [6]. Although intramedullary hemangioblastoma is characteristic of vHL, it also occurs sporadically [1] and overall represents the third most common spinal intramedullary neoplasm [7], with an incidence of 1.5–15% [8,9]. As SIMH characteristically do not undergo malignant transformation, the first-line treatment strategy is a gross total resection to allow neuronal decompression in case of neurological deficit [10,11]. However, so far, it remains unclear if surgical treatment is beneficial for all patients and whether preventive surgery is indicated in asymptomatic cases or when patients present with mild deficits.

In recent years, quality indicators (QI) for medical services and procedures have been established, allowing a better assessment of the postoperative outcome and identification of potential risk factors that impact the success of surgical treatment strategies. These quality indicators include 90-day readmission, reoperation, mortality rates, length of hospital stay (LOS), rates of nosocomial infections and surgical site infections (SSI) as well as quality-of-life (QoL) assessment [12,13,14,15]. QoL assessment after surgery has helped to predict potential long-term outcomes for other tumor entities, such as spinal meningiomas [16] or ependymomas [17], and has significantly improved patient care, allowing the weighing of a surgical strategy against conservative treatment.

As patients with spinal hemangioblastoma in the context of vHL usually suffer from multiple tumors in the CNS and therefore often undergo multiple surgeries, a careful weighing of chances and risks of an additional spinal surgical intervention is essential to achieve the best long-term outcome for this patient group.

Therefore, the aim of this study was to evaluate QI and long-term QoL in patients who underwent intramedullary hemangioblastoma resection at our department of neurosurgery to investigate potential risk factors for impaired outcomes and characterize patients who benefit from surgical intervention.

## 2. Materials and Methods

### 2.1. Study Design

All patients who underwent intramedullary hemangioblastoma resection between 2014 and 2022 at our neurosurgical department were included. The following data of the hospital electronic records were analyzed: age, sex, human genetic data regarding von Hippel-Lindau disease, tumor size, the presence of syrinx, surgical approach, the extent of resection (EOR), use of preoperative embolization, use of intraoperative neuromonitoring (IOM) and intraoperative indocyanine-green (ICG) angiography, blood loss (ml), duration of surgery (minutes), length of hospital Stay (LOS; days), 90-day nosocomial infections, 90-day surgical site infection, unplanned 90-day readmission and 90-day reoperation, 90-day mortality and tumor progress or recurrence. The study was conducted in accordance with the Declaration of Helsinki and approved by the Institutional Review Board (or Ethics Committee) of the University of Münster, Germany (reference number 2021-714-f-S, 15 February 2022).

### 2.2. Surgical Intervention

Surgical resection was performed via dorsal or dorso-lateral approach using laminoplasty or hemilaminectomy according to the tumor’s location in the spinal cord. During surgery, intra-operative neurophysiological monitoring (IOM, inomed Medizintechnik GmbH, Emmendingen, Germany) was conducted, including motor-evoked potentials (MEP), sensory-evoked potentials (SEP) of upper and lower extremities, and D-wave. The need for embolization was discussed in an interdisciplinary preoperative case conference; the main criteria for an embolization were larger tumor size and technical feasibility. After the exposure of the dura, intraoperative sonography was performed to detect the tumor’s location and the dura was opened. To visualize and coagulate the tumor, feeding vasculature indocyanine-green (ICG) angiography (Zeiss Pentero or Kinevo, Carl Zeiss, Oberkochen, Germany) was applied. Consequently, the tumor was resected using blunt dissection and non-stick bipolar forceps (Spetzler-Malis^®^, Stryker, Kalamazoo, MI, USA). In those cases where the extent of performed resection was unclear, ICG angiography was repeated. In the end, the dura was closed using a 6-0 monofil continuous suture. For those cases where laminoplasty was selected as a surgical approach, the lamina was fixated with mini-plates. The extent of resection was evaluated early using an MRI, within 48 h after surgery.

### 2.3. Outcome and Assessment of QoL

Neurological patients’ status was measured according to the modified McCormick Scale [16,17], ranging from I (no symptoms or minimal dysesthesia) to V (paraplegic/quadriplegic). The status was evaluated both before and after surgery independently by two of the authors (MS and SR). For further dichotomic calculations, the McCormick scale I was considered as “favorable”, and II to V as “unfavorable”.

QoL was assessed by the EQ-5D-3L questionnaire [18,19]. It contains five questions on mobility, self-care, activity, pain and anxiety. For each question, three responses were available, each ranging from 1 (no difficulties), over 2 (moderate limitations), to 3 (major limitations). In addition, patients rated their overall well-being on a scale of 0–100 (EQ-VAS).

### 2.4. Statistical Analysis

A statistical analysis was performed using SPSS Statistics 29.0 (IBM Corp., Armonk, NY, USA). Categorical variables are shown as absolute and relative frequencies. Parametric values are presented in mean and standard deviation (SD). Non-parametric values are presented as the median and interquartile range (IQR, 25% quartile and 75% quartile). Fisher’s exact test was performed to compare groups of binary categorical variables. A two-tailed Student *t*-test was used as a parametric and a two-sided Mann–Whitney U-test (MWU) as a non-parametric test. A preoperative semiautomatic tumor volumetric analysis via segmentation was performed using Brainlab elements^®^ software (Brainlab AG, Munich, Germany) and expressed in milliliters (cm^3^). A probability value less than *p* < 0.05 was considered statistically significant.

## 3. Results

### 3.1. Population

At our department of neurosurgery, 23 (14 male and 9 female) patients with intramedullary hemangioblastoma were treated between 2014 and 2022. Four patients underwent surgery twice at different locations due to multiple tumors. Patients’ ages ranged between 15 and 63 years, with a mean age of 38 (±13.17) years. Seventy-eight percent (18/23) were diagnosed with von Hippel-Lindau disease. All four patients who underwent surgery twice or more had a von Hippel-Lindau mutation. Forty-eight percent of the tumors were located (13/27) within the cervical spinal cord, while the other forty-eight percent (13/27) fell in the thoracic level. One patient had a tumor at the level of L-1.

In 74.1% of the cases (20/27), the tumor was associated with syringomyelia. The median tumor volume was 0.22 cm^3^ (±0.128), ranging between 0.06 cm^3^ to 2.6 cm^3^. Neither tumor volume nor the presence of syringomyelia had any significant impact on the neurological status. Figure 1, Figure 2 and Figure 3 demonstrate example cases. See Table 1 for further patients’ characteristics.

### 3.2. Surgical Intervention

Preoperative embolization was performed in three cases (tumor volumes: 1.4 cm^3^, 2.6 cm^3^ and 0.67 cm^3^). Hemilaminectomy was the main technique performed in 66.7% (18/27) of cases, followed by laminoplasty in 29.6% (8/27). Laminectomy was carried out only once, for the resection of a tumor at the level of C-1 (3.7%).

Gross total resection (GTR) was achieved in 81.5% of the cases (22/27). In five patients, only subtotal resection (STR) could be performed due to a concern regarding neurological worsening due to the deterioration of IOM findings. The mean duration of the surgery was 310 min (±159.98). The average amount of blood loss was 200 mL (±562.8), ranging from negligible amount in eight cases to 2500 mL in one single case. The median length of hospital (LOS) stay was six days (IQR: 5–7). See Table 2 for further data regarding surgery and postoperative outcomes.

### 3.3. Neurological Outcome and QI

Long term follow-up and clinical courses were available to all patients but one. In two cases (7.41%) perioperative nosocomial urinary tract infections (UTI) were reported. Reoperations within the first 90 days after index surgery were performed due to one case of residual tumor and one case of cerebrospinal fluid leakage (CSFL). One additional patient was readmitted within 30 days due to a superficial wound infection, which was treated non-surgically.

Postoperatively, neurological improvement of one point on the McCormick scale was reported in 25.9% (7/27) of cases. However, in 74.1% (20/27) of the cases, the neurological status was described as unchanged. Out of the total patient cohort, 10 individuals (constituting 37.04%) initially exhibited a preoperative McCormick scale of I, and therefore their neurological status could not be further ameliorated through the intervention. Patients with a preoperative McCormick scale II did not experience any deterioration (9/9) and even displayed an improved neurological outcome in three cases (3/9, 33.33%) after surgery. In contrast, of the two patients with a preoperative McCormick scale of III, one (50%) showed an improvement of one point and the other a deterioration of one point. Disabled Among the six patients that showed a preoperative McCormick scale of IV, a pair of individuals (constituting 33.33% within this subset) exhibited an improvement of one point. Conversely, the remaining four patients (making up 66.66% of the subset) did not manifest any amelioration in neurological status, resulting in the enduring presence of substantial disability (Figure 4).

To conduct a more comprehensive analysis of these findings, the cohort of patients was stratified into two distinct categories. The first group encompassed patients exhibiting a “favorable outcome,” as defined by a postoperative McCormick scale score of I. The second group comprised patients experiencing an “unfavorable outcome,” as characterized by postoperative McCormick scale scores ranging from II to V. This analysis demonstrated that the postoperative neurological status had a large influence on the favorable neurological outcome (*p* < 0.001). Patients with no or mild symptoms (McCormick scale I and II) had significantly more favorable outcomes than patients with more severe neurological deficits (McCormick scale III and IV, *p* = 0.001). Other variables that influenced neurological outcomes were the presence of vHL mutations (*p* = 0.010) and STR (*p* = 0.017).

Not surprising is the finding that patients with unfavorable outcomes stayed longer in the hospital with a mean LOS of 8.29 days in comparison to 5.85 days in case of a favorable outcome (*p* = 0.022). Complications were also more common in the unfavorable group, however, without statistical significance. See Table 3 for further data regarding these findings.

### 3.4. Quality of Life

In addition to the neurological outcome, we intended to evaluate the quality of life (QoL) of patients after resection of SIMH, for that reason we contacted all patients treated and received a total of 14 completed EQ-5D-3L questionnaires (60,9%); no feedback could be obtained from patients who meanwhile died or changed contact information due to movement/new phone number. In further evaluation, we compared the results of these questionnaires to the post-operative neurological status. Patients with a better neurological outcome (McCormick Scale of I–II, 11 patients) ranked their general quality of life (EQ_VAS) above 75–100 points in 81.8% of cases (N = 9/11) and 45.5% of them (N = 5) rated their general quality of life with ≥90 points. In comparison, patients with higher McCormick scales after surgery (McCormick Scale III-IV, 3 patients) rated the overall quality of life with 25–30 points in two out of three cases (66.7%). Although these results showed a significant association (*p* = 0.038), we could not confirm this association with the outcome groups mentioned above (*p* = 0.259).

Overall, there were three exceptions; two patients from the favorable group rated their general life satisfaction with 45 and 30 points, respectively, while one patient from the unfavorable group, with a postoperative McCormick Score of IV, rated his subjective satisfaction with 70 points.

In addition to the EQ VAS, patients evaluated their mobility, self-care, activity, pain and anxiety. Most complaints were regarding pain, followed by anxiety and impaired activity. Most patients did not report difficulties in mobility and self-care. Patient 9, who has a good neurological status (McCormick scale I) and reported his general satisfaction with 30 points, also reported severe pain and anxiety (Table 4).

## 4. Discussion

Spinal intramedullary hemangioblastomas (SIMH) are characteristically found in vHL patients but also occur sporadically in otherwise healthy adults [1]. Clinical presentation is associated with the intramedullary location and patients usually display pain and/or sensomotor deficits [5] but are also often found coincidentally in asymptomatic vHL patients. As those tumors have not been observed to show malignant transformation, it remains unclear whether patients should undergo surgery when they display no or mild neurological deficits. Therefore, we investigated the neurological outcome and the quality of life following SIMH resection in a single-center case series.

### 4.1. Favorable Postoperative Outcome May Depend on the Preoperative Neurological Status

Our results indicate that the postoperative status following SIMH resection may depend mainly on the preoperative neurological status of the patients. Patients with no or mild symptoms who were classified preoperatively as McCormick Scale I or II either did not show new neurological deficits or even improved their neurological status postoperatively. Patients who already showed McCormick scale III or higher prior to intervention did improve in some cases, however, still had significant neurological deficits after surgery. In one case we even noticed a further neurological deterioration. These results are in line with previously published studies investigating the outcome after SIMH and other intradural tumor resections [10,11,20,21,22] and highlight that severe neuronal damage would not fully recover after surgery. However, in a recent large case series, the authors had some evidence of this association but could not prove statistical significance [23].

Moreover, the results of this study suggest that the extent of resection (EOR) is higher in the case of mild neurological disorders in comparison to the case when more severe neurological deficits are preexisting, again showing the vulnerability of the spinal cord in that case. In all cases of STR in this series, surgery had to be prematurely terminated due to deterioration of IOM potentials during tumor resection [23,24,25]. In addition, even if not statistically significant in the results, patients with more severe deficits had more postoperative complications, higher blood loss and longer duration of the surgery. This suggests that resection of SIMH in an early stage of disease is safe, especially when IOM is utilized. On the other hand, surgery in case of severe neurological deficits bears a higher risk for complications and incomplete resection, and patients would be stabilized or improve slightly, however, full recovery is rather unlikely.

These results indicate that surgical treatment of SIMH should be conducted in an early stage of the disease before severe neurological deficits occur. Since even in the case of mild neurological deficits a full recovery cannot be guaranteed, a strategy of wait and see, which is often propagated, seems to be less recommended. Butenschoen and colleagues demonstrated also that a longer duration of symptoms is associated with worse neurological outcomes [10,11,22].

Another risk factor for unfavorable outcomes was the presence of vHL mutations. Again, Feletti and colleagues suspected this too, but could not confirm this in their results [23]. vHL patients usually also display other tumors in the CNS and outside of it such as in the eyes, kidney, and pancreas [2], moreover, they may have more than one SIMH. For this reason, they usually undergo multiple surgeries, which is accompanied by an enormous burden on the patients and therefore requires careful consideration as to whether further surgery is beneficial. Consequently, the assessment of quality of life to evaluate the postoperative outcome appears particularly important in this population. According to the results of our study and in accordance with previous studies we recommend early surgery also in case of vHL [10,11,22].

### 4.2. A Good Postoperative Outcome Is Associated with a Higher Quality of Life

As expected intuitively, we observed an association between quality of life and postoperative neurological outcome. Patients who showed a better postoperative outcome ranked their life quality significantly higher than those patients who already suffered from more severe neurological deficits and showed a worse postoperative outcome. These results are similar to those after the resection of other intradural tumors [20].

Interestingly the overall quality of life reported by patients was good, and patients most frequently complained about moderate pain, followed by moderately reduced activity and anxiety, suggesting a mandatory adjuvant treatment of these symptoms. Several studies have highlighted that supportive care of oncological patients who suffer from neurological symptoms reduces the rate of psychological disorders, pain, and anxiety [26,27,28,29].

### 4.3. An Interdisciplinary Approach Prevents Complications

Overall, we had two revision surgeries within 90 days, one due to a residual tumor in postoperative imaging, which was not visible in preoperative imaging and during surgery. In the second case, we observed CSFL after surgery. Furthermore, one patient was readmitted within 90 days due to a superficial surgical site infection which was treated conservatively. In addition to that, two patients showed a nosocomial UTI following Foley catheter placement during surgery.

The overall positive surgical outcome in this study was very likely associated with the interdisciplinary treatment approach. If needed, interventional radiologists performed preoperative tumor embolization minimizing the risk for extensive intraoperative bleeding which was only reported in one case without preoperative embolization. Furthermore, GTR could be achieved in most cases via a minimal invasive unilateral approach and hemilaminectomy. Hemilaminectomy with less muscle detachment results in less pain, shorter recovery, and shorter LOS [30,31]. In other cases, laminoplasty was performed, and the laminae were fixed after tumor resection. Furthermore, both hemilaminectomy and laminoplasty reduce the risk of CSF leak [32]. Indeed, the only patient in this series with a CSFL had a laminectomy at the level of C1. Previous publications demonstrated very good outcomes after a minimal-invasive, unilateral approach [22,23,33].

Moreover, the intraoperative application of ICG angiography was as previously described [34,35,36] considerably helpful in visualizing the tumor vascular architecture intraoperatively prior to resection and in some cases even guided the surgeon to the tumor when it was relatively small. Finally, the utility of IOM gave the surgeon security to achieve GTR while potential remained stable during surgery and prevented potential neuronal damage when potentials showed significant deterioration during surgery [24,25].

### 4.4. Limitations

The study’s main limitation is the retrospective nature of the analysis and the relatively small number of patients due to the rare incidence of these tumors.

## 5. Conclusions

Resection of intramedullary hemangioblastoma in patients with no, or minor symptoms is associated with a favorable neurological outcome and overall higher quality of life. Therefore, we recommend performing surgery prior to the development of significant neurological disorders. Prospective and larger-scale studies and patients’ registries may offer better results in the future.

## Figures and Tables

**Figure 1 medicina-59-01611-f001:**
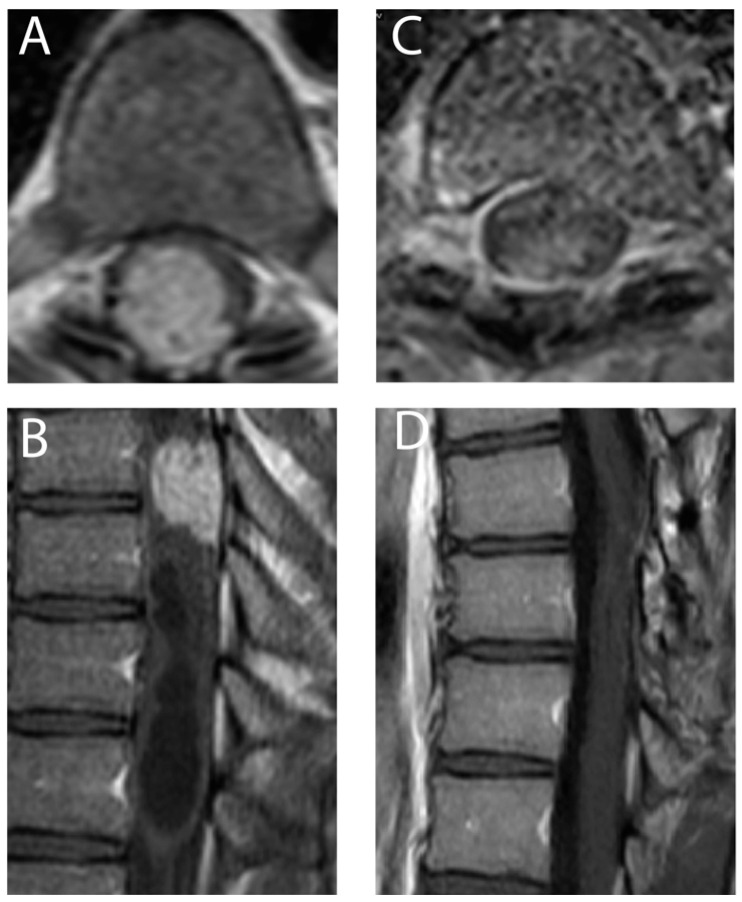
Axial (**A**) and sagittal (**B**) T1 weighted MRI with gadolinium images demonstrate a large spinal intramedullary hemangioblastoma in the thoracic spine with syringomyelia. This patient had only mild dysesthesia in the lower extremities, without motor deficit or gait disturbance (McCormick scale I). After gross total resection, demonstrated also in axial (**C**) and sagittal (**D**) T1 weighted MRI with gadolinium, the patient recovered very quickly and had an excellent neurological outcome (McCormick scale I).

**Figure 2 medicina-59-01611-f002:**
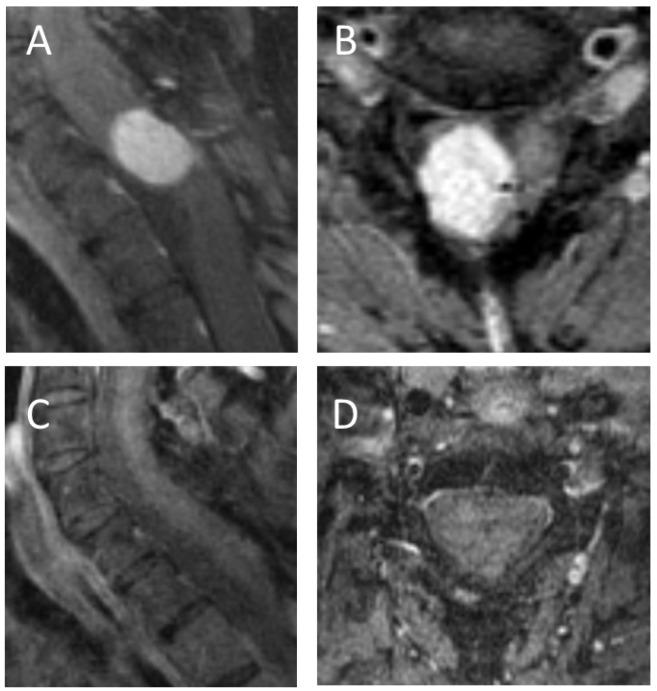
Sagittal (**A**) and axial (**B**) T1 weighted MRI with gadolinium images of a patient with a large spinal intramedullary hemangioblastoma in the cervical spine; preoperatively this patient had severe neurological disorders with motor weakness and ataxia (McCormick scale IV), although a gross total resection was performed, as shown on postoperative sagittal (**C**) and axial (**D**) T1 weighted images with gadolinium, the neurology did not recover significantly, although the motor weakness partially recovered the patient still has a severe gait ataxia.

**Figure 3 medicina-59-01611-f003:**
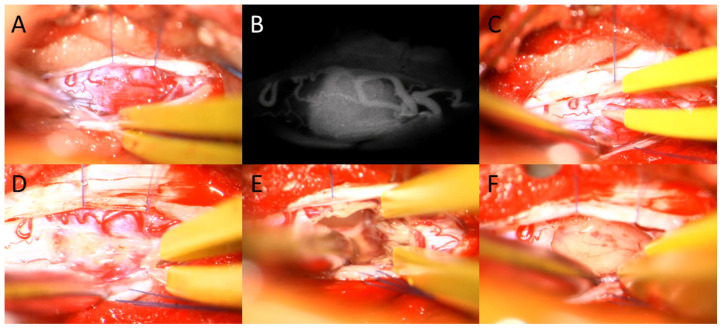
Intraoperative images of microsurgical resection of an intramedullary hemangioblastoma. After performing a unilateral hemilaminectomy and durotomy the hemangioblastoma can be located (**A**), notice the vascular structures. With the help of Indocyanine green (ICG) angiography, the feeder artery can be detected (**B**), and subsequently coagulated, as shown in image (**C**). Afterwards, the tumor can be resected as demonstrated in images (**D**,**E**). Image (**F**) shows the spinal cord and tumor cavity after gross total resection.

**Figure 4 medicina-59-01611-f004:**
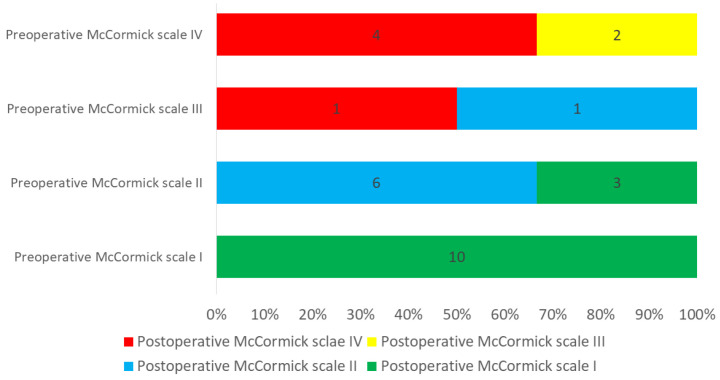
Neurological outcome rated on McCormick Scale comparing pre- and postoperative neurosurgical status. The Y axis represents the preoperative neurological status according to the McCormick scale and the X axis denotes the number and distribution of patients within each respective scale category. The color differentiation serves to visually depict postoperative neurological status (green: McCormick I, blue: McCormick II, yellow: McCormick III and red: McCormick IV). Among the patients initially assessed with a preoperative McCormick scale of I, the entirety of this subset exhibited an unaltered status at that level (100%). In comparison, a third (N = 3) of the patients with preoperative McCormick scale of II improved their neurological status, while the remaining two-thirds (N = 6) maintained their preoperative status. Out of the patients with preoperative McCormick scale of III, one patient deteriorated to scale IV whereas the other improved to scale II. Patients with preoperative McCormick scale IV improved to scale III in one-third (N = 2) of the cases, while the remaining two-thirds sustained their status (N = 4).

**Table 1 medicina-59-01611-t001:** Patients‘ characteristics: (SD: standard deviation).

Variable	Value
**Age (Mean** **±** **SD)**	38.20 ± 13.12
**Sex female/male (N/%)**	11 (40.74%)/16 (59.26%)
**Localization**	
Cervical (N/%)	13 (48.15%)
Thoracic	13 (48.15%)
Lumbar	1 (3.70%)
**Tumor volume (cm^3^, Mean** **±** **SD)**	0.60 ± 0.93
**Preoperative neurological status**	
McCormick I (N/%)	10 (37.04%)
McCormick II (N/%)	9 (33.33%)
McCormick III (N/%)	2 (7.41%)
Mc Cormick IV (N/%)	6 (22.22%)
**Syringomyelia (N/%)**	20 (74.07%)
**Von Hippel-Lindau mutation (N/%)**	22 (81.48%)

**Table 2 medicina-59-01611-t002:** Surgical data and postoperative outcomes: (SD = standard deviation).

Variable	Value
**Surgical approach**	
Laminoplasty (N/%)	8 (29.63%)
Unilateral laminotomy	18 (66.7%)
Laminectomy	1 (3.7%)
**Dura closure**	
Direct closure	23 (85.19%)
Extension duroplasty	4 (14.81%)
**Extent of resection (EOR)**	
Gros total resection (GTR)	22 (81.48%)
Subtotal resection (STR)	5 (18.52%)
**Blood loss (ml, mean,** **±** **SD)**	345.19 ± 562.87
**Preoperative embolization (N/%)**	3 (11.11%)
**Indocyanine green (ICG) angiography (N/%)**	22 (81.48%)
**Intraoperative monitoring (N/%)**	27 (100%)
**Duration of surgery (Minutes, mean,** **±** **SD)**	332.3 ± 159.98
**Postoperative neurological status**	
McCormick I (N/%)	13 (48.15%)
McCormick II (N/%)	7 (25.93%)
McCormick III (N/%)	2 (7.41%)
McCormick IV (N/%)	5 (18.52%)
**Length of hospital stay (days, mean** **± SD** **)**	7.11 ± 3.61
**Nosocomial infection**	
Urinary tract infection	2 (7.41%)
**90 days readmission (N/%)**	3 (11.11%)
**90 days re-operation (N/%)**	2 (7.41%)

**Table 3 medicina-59-01611-t003:** Patients’ characteristics stratified according to postoperative neurological status (SD = standard deviation). The favorable outcome group includes patients with a postoperative McCormick scale I and the unfavorable group patients with a postoperative McCormick scale II to IV.

	Favorable Outcome (N = 13)	Unfavorable Outcome (N = 14)	*p*
**Sex (females, N/%)**	5 (38.46%)	6 (42.86%)	0.816
**Age (years, mean, ± SD)**	35.46 ±10.10	42.00 ±15.39	0.302
**Tumor volume (ml)**	0.46 ± 0.72	0.71 ± 1.08	0.720
**Localization**			0.568
Cervical	7	6	
Thoraco-lumbar	6	8	
**Von Hippel-Lindau mutation (N/%)**	8 (61.53%)	14 (100%)	0.010
**Syringomyelia (N/%)**	3 (23.08%)	4 (28.57%)	0.745
**Preoperative neurological status (N/%)**			<0.001
McCormick scale 1	10 (76.92%)	0	
McCormick scale 2	3 (23.08%)	6 (42.86%)	
McCormick scale 3	0	2 (14.29%)	
McCormick scale 4	0	6 (42.86%)	
**Preoperative embolization (N/%)**	1 (7.7.69%)	2 (14.29%)	0.586
**Surgical approach**			0.948
Hemilaminectomy	9 (69.23%)	9 (64.29%)	
Laminectomy	1 (7.69%)	0	
Laminoplasty	3 (23.08%)	5 (35.71%)	
**Primary surgery (N/%)**	12 (92.3%)	12 (85.71%)	0.586
**Indocyanine green (ICG)** **angiography (N/%)**	12 (92.31%)	10 (71.43%)	0.163
**Intraoperative neuromonitoring**	13 (100%)	14 (100%)	>0.999
**Duration of surgery (minutes, mean, ± SD)**	273.23 ± 76.42	387.14 ± 197.66	0.185
**Blood loss (ml, mean, ± SD)**	288.46 ± 445.42	430.00 ± 646.04	0.458
**Gross total resection (N/%)**	13 (100%)	9 (64.29%)	0.017
**Expansion duraplasty (N/%)**	1 (7.69%)	3 (21.43%)	0.315
**Bed rest after surgery (N/%)**	7 (53.85%)	7 (50.00%)	0.842
**Leakage of cerebrospinal fluid (N/%)**	0	1 (7.14%)	0.326
**Length of hospital stay (days, mean, ± SD)**	5.85 ± 2.37	8.29 ± 4.21	0.022
**Nosocomial infections (N/%)**	0	2 (14.29%)	0.157
**Readmission within 90 days (N/%)**	1 (7.69%)	2 (14.29%)	0.586
**Re-Surgery within 90 days (N/%)**	0	2 (14.29%)	0.157
**Tumor recurrence (N/%)**	0	0	>0.999

**Table 4 medicina-59-01611-t004:** Assessment of the quality-of-life according to EQ-5D questionnaire.

Patient	Mobility	Self-Care	Activity	Pain	Anxiety	EQ_VAS	McCormick Scale Pre-Operative	McCormick Scale Post-Operative
**1**	1	1	1	1	2	85	1	1
**2**	1	1	1	2	1	85	2	2
**3**	2	1	2	2	1	75	3	2
**4**	2	2	2	2	2	40	4	4
**5**	1	1	2	2	1	70	3	4
**6**	1	1	2	2	2	85	2	2
**7**	2	1	2	2	2	45	2	1
**8**	1	1	1	1	2	90	2	2
**9**	2	1	2	3	3	30	1	1
**10**	2	2	2	3	2	25	4	4
**11**	1	1	1	2	1	90	2	1
**12**	1	1	1	1	2	90	1	1
**13**	1	1	2	2	1	90	1	1
**14**	1	1	1	2	1	92	1	1
**Median**	1	1	2	2	2	85		
**IQR**	1–2	1	1–2	1.5–2	1–2	43.75–90		

## Data Availability

Study data can be obtained by contacting the corresponding author, michael.schwake@ukmuenster.de.

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
