# Peer review of "Resection of Intramedullary Hemangioblastoma: Timing of Surgery and Its Impact on Neurological Outcome and Quality of Life"

_medicina, 2023, doi:10.3390/medicina59091611_

Round 1
Reviewer 1 Report
This is a globally well written paper that retrospectively describes a series of surgically resected intramedullary hemangioblastomas. The authors found that the postoperative outcome mainly depend on the pre-operative status and this suggest the importance of offering an early treatment before deterioration occurs. They also explored quality of life of part of these patients. I believe that they should clearly improve the bibliography. As an example i am surprised that they did not cite and comment the work by Feletti et al. that recently published a large series of spinal hemangioblastomas with similar conclusions.
Legend of figure I should be corrected adding the appropriate labels.
The authors should modify figure 2 adding also the post-operative results. I also encourage them to add a figure with some nice intraoperative photos of one of their case.
Author Response
This is a globally well written paper that retrospectively describes a series of surgically resected intramedullary hemangioblastomas. The authors found that the postoperative outcome mainly depend on the pre-operative status and this suggest the importance of offering an early treatment before deterioration occurs. They also explored quality of life of part of these patients.
We thank reviewer 1 for his/her comments. Indeed, our conclusion is that spinal hemangioblastomas should be resected before major neurological deficits, such as motor deficits of ataxia occur.
- I believe that they should clearly improve the bibliography. As an example i am surprised that they did not cite and comment the work by Feletti et al. that recently published a large series of spinal hemangioblastomas with similar conclusions.
Thank you for this recommendation, we added the paper of Feletti et al., and further publication of Butenschoen et al., Harati et al., and Krüger et al.
- Legend of figure I should be corrected adding the appropriate labels.
We corrected the labels of all figures.
- The authors should modify figure 2 adding also the post-operative results. I also encourage them to add a figure with some nice intraoperative photos of one of their case.
We added the postoperative images, and nice intraoperative photos (Figure 3)
Reviewer 2 Report
Dear Authors, I appreciate your effort writing an interesting article regarding surgical timing of of the intramedullary hemangioblastoma resection, its neurological outcomes and quality of life. The paper is well written and covers an interesting topic. However, there is a critical point that need to be addressed before decision.
Authors conclude that resection surgery is recommended prior to the development of significant neurological disorders by comparing patients with favorable neurological status (McCormick Scale I or II) and those with unfavorable status (McCormick Scale III to V). I think that there is a significant error in methodology. It seems to be unreasonable to compare two groups with different start point and conclude the preoperative favorable neurological status as a predicting factor for favorable outcomes. Patients with preoperative favorable neurological status had good neurological outcomes, and those with preoperative unfavorable status had relatively poor neurological outcomes. It looks too obvious. What would be the result of not performing surgery? I think that the results would be same if these patients did not have surgery. Instead, to identify the predicting factors for good outcomes or risk factors for poor outcomes, it would be appropriate to divide the cohort into two groups with good outcomes and poor outcomes, and compare and analyze them using the logistic regression analysis.
Author Response
Dear Authors, I appreciate your effort writing an interesting article regarding surgical timing of of the intramedullary hemangioblastoma resection, its neurological outcomes and quality of life. The paper is well written and covers an interesting topic. However, there is a critical point that need to be addressed before decision.
- Authors conclude that resection surgery is recommended prior to the development of significant neurological disorders by comparing patients with favorable neurological status (McCormick Scale I or II) and those with unfavorable status (McCormick Scale III to V). I think that there is a significant error in methodology. It seems to be unreasonable to compare two groups with different start point and conclude the preoperative favorable neurological status as a predicting factor for favorable outcomes. Patients with preoperative favorable neurological status had good neurological outcomes, and those with preoperative unfavorable status had relatively poor neurological outcomes. It looks too obvious. What would be the result of not performing surgery? I think that the results would be same if these patients did not have surgery.
We thank reviewer 2 for his comment, of course it is impossible to perform the perfect allocation of the patients, as the study was done in retrospective fashion. Nor do we know why some patients were presented so late in our department. A prospective trial comparing early surgery to late surgery (neurological deterioration after “wait and see”) would surly give a much better scientific methodology.
However, we assume that many patients with worse neurological conditions prior surgery in this study, especially patients with vHL, were in a such “wait and see” regime. Indeed, according to our experience many surgeons and neurologists don’t recommend surgery in these cases, assuming resection is not required and would only cause unnecessary harm to the asymptomatic patients. Indeed, resection of intramedullary tumors is one of the most complex procedures in neurosurgery. On the other hand, according to our results such a regime is not the best option for patients, and tumor resection should be performed in an early stage of the disease. Moreover, in this stage surgery seems to be quite safe, especially when the surgeons use IOM, ICG angiography and minimal-invasive approaches. In contrary in case of more severe neurological deficits, such as ataxia or motor weakness, the patients recovered only in about third of the cases, and in most cases remained quite disabled.
- Instead, to identify the predicting factors for good outcomes or risk factors for poor outcomes, it would be appropriate to divide the cohort into two groups with good outcomes and poor outcomes, and compare and analyze them using the logistic regression analysis.
We followed the recommendation of reviewer 2 and present the results divided into two groups. The first, we named “favorable outcome”, includes all patients with a postoperative McCormick scale 1 and the second, “unfavorable outcome”, includes patients with worse neurological outcome, McCormick scale > 1. The results are presented in the new table 3. Here we could find out, that the main risk factors for unfavorable outcome are a preexisting neurological deficit and vHL mutation. These results are in concurrence with previous studied on spinal tumors, including metastases, juxtamedullary, and intramedullary tumors.
Round 2
Reviewer 2 Report
I appreciate your hard work and dedication writing an interesting paper and answering my questions clearly. In addition, I'm grateful for a quick revision in methodology. I deeply agree with the authors' conclusion that early surgical intervention before the development of neurological deficits may offer better outcomes and quality of life.